# Macrophage-like Cells Are Increased in Retinal Vein Occlusion and Correlate with More Intravitreal Injections and Worse Visual Acuity Outcomes

**DOI:** 10.3390/jpm13010045

**Published:** 2022-12-26

**Authors:** Sean M. Rangwani, Stephen Hawn, Nathan C. Sklar, Rukhsana G. Mirza, Jeremy A. Lavine

**Affiliations:** Department of Ophthalmology, Feinberg School of Medicine, Northwestern University, Chicago, IL 60611, USA

**Keywords:** optical coherence tomography angiography, OCTA, macula, retina, macrophage-like cell (MLC), retinal vein occlusion, RVO

## Abstract

Macrophage-like cells (MLCs) are an emerging retinal biomarker. MLCs are increased in retinal vein occlusion (RVO) eyes, but their predictive value is unknown. This study investigated if MLCs can predict meaningful clinical outcomes. This prospective, cross-sectional study involved 46 eyes from 23 patients with unilateral RVO. Patients’ unaffected eyes were used as matched controls. MLCs were quantified to determine MLC density and percent image area. We collected demographic, clinical, ocular, and imaging characteristics at the time of MLC imaging. We additionally recorded best corrected visual acuity (BCVA) and number of intravitreal injections at 6 months and 12 months post-imaging. MLC density and percent area increased by 1.86 (*p* = 0.0266)- and 1.94 (*p* = 0.0415)-fold in RVO compared to control eyes. We found no significant correlation between MLC parameters and any baseline characteristic. MLC density was positively correlated with the number of intravitreal injections at 6 months (*n* = 12, *r* = 0.62, *p* = 0.03) and 12 months (*n* = 9, *r* = 0.80, *p* = 0.009) post-imaging. MLC percent area was correlated with LogMAR BCVA change over 12 months (*n* = 17, *r* = 0.57, *p* = 0.02). High MLC counts correlated with more future intravitreal injections and worse visual acuity outcomes, suggesting that MLCs are a biomarker for treatment resistant RVO eyes.

## 1. Introduction

Retinal vein occlusion (RVO) is the second most common retinovascular disorder behind diabetic retinopathy (DR), and a significant cause of vision loss [1]. Loss of visual acuity in RVO patients occurs through two mechanisms: ischemia and macular edema. Occlusion of the venous outflow damages the vasculature and causes ischemia, which can directly lead to vision loss due to poor retinal perfusion. Additionally, tissue ischemia triggers inflammation, increased vascular permeability, and vision-threatening macular edema [2]. In RVO patients, the vitreous and aqueous humor show high levels of inflammatory and angiogenic mediators, including interleukins, interferons, tumor necrosis factor, and vascular endothelial growth factor (VEGF) [3,4,5,6]. As a result, the two most common treatments for RVO include intravitreal anti-VEGF and steroid medications [7,8,9,10]. Anti-VEGF medications are first line therapies [11]; in patients who respond poorly to anti-VEGF treatments, steroids are effective but have higher side effect profiles [12,13]. Therefore, biomarkers are an important area of investigation to determine which patients may benefit from anti-inflammatory therapies.

Macrophage-like cells (MLCs) are an emerging potential inflammatory biomarker in retinal vascular disease. Using either optical coherence tomography angiography (OCTA) or adaptive optics scanning laser ophthalmoscopy (AO-SLO) imaging, cells which are mobile and ramified are detectable on the surface of the retina [14,15,16]. These cells are believed to be macrophages based upon their ramified appearance, mobility, and expression of *Cx3cr1* (a pan-macrophage marker) in mice using AO-SLO [17]. Furthermore, we recently showed that OCTA imaging can also detect MLCs in mice [18]. MLCs include microglia, perivascular macrophages, and vitreal hyalocytes at steady state, and also pro-inflammatory cells like monocytes, monocyte-derived macrophages, and neutrophils during neuroinflammation [18]. In mice, experimental RVO causes microglia activation and monocyte-derived macrophage accumulation [19]. These data suggest that macrophages are important in the pathogenesis of RVO, and MLC imaging could be an inflammatory biomarker for RVO patients.

We previously showed that MLC density was significantly greater in proliferative DR (PDR) eyes compared to control, diabetes mellitus without DR, and non-PDR (NPDR) eyes [20]. Since PDR shares many pathophysiologic features with RVO, including ischemia and inflammation, we hypothesized that MLCs are more numerous in RVO eyes. In agreement, Zeng et al. and Wang et al. recently showed that MLC density was significantly increased in RVO eyes [21,22]. However, Zeng et al. did not investigate if MLCs can predict meaningful RVO outcomes. We found that MLC density and percent area were significantly higher in RVO eyes compared to control eyes. Furthermore, MLC metrics correlated with the number of intravitreal injections and BCVA at 12 months post-imaging. These data demonstrate that MLCs are increased in RVO eyes, patients with less MLCs require less injections, and patients with less MLCs have better visual acuity outcomes. 

## 2. Materials and Methods

### 2.1. Subjects

This prospective, cross-sectional study enrolled 46 eyes from 23 patients with RVO seen at the Department of Ophthalmology at Northwestern University in Chicago, Illinois from May 2020 to November 2021. This study was approved by the Institutional Review Board of Northwestern University and was conducted in accordance with the tenets of the Declaration of Helsinki and regulations of the Health Insurance Portability and Accountability Act. All subjects provided written informed consent before participation. 

Inclusion criteria were age ≥ 18 years and the diagnosis of unilateral RVO. Patients were excluded if they had any additional retinal disease in either eye, significant media or lens opacities that would obscure images, or vitreoretinal pathology including epiretinal membrane or vitreomacular traction/adhesion. The non-RVO eye was used as a control. At their initial visits, all subjects had a complete ocular exam, which included best corrected visual acuity (BCVA), slit lamp examination, and OCT imaging. From the electronic health records system, we collected demographic information (age, gender), clinical characteristics (diabetes, hypertension, RVO type, number of intravitreal injections), and ophthalmic clinical examination (BCVA, OCT central subfield thickness [CST]) at the time of MLC imaging. We additionally collected BCVA and number of intravitreal injections (all anti-VEGF in our cohort) at 6 months and 12 months post-imaging. 

### 2.2. OCTA Imaging and Processing

The PLEX Elite 9000 (Carl Zeiss Meditec Inc., Dublin, CA, USA) was used to obtain a single 3 × 3 mm and 6 × 6 mm scan of the retina centered on the fovea of the RVO and control eye of each subject. We segmented each scan to capture the superficial vascular plexus (SVP), retinal nerve fiber layer (RNFL, 0–27 microns below the internal limiting membrane [ILM]), and MLC slab (3–6 microns above the ILM). We and Castanos et al. previously used 0–3 microns above the ILM, however, on the PLEX Elite 9000, a 3-micron slab 3 microns above the ILM captured equal MLC number with reduced artifact from the RNFL layer [14,20].

Using FIJI software, a distribution of the program ImageJ (National Institutes of Health, Bethesda, MD, USA), we registered the 3 × 3 mm and 6 × 6 mm SVP slabs using the Register Virtual Stack Slices Plugin (Featured Extraction Model = Rigid, Registration Model = Elastic), as previously described [20]. We next applied the saved transformation matrix from the SVP slab to the RNFL and MLC slab using the Transform Virtual Stack Slices Plugin. We averaged the registered SVP, RNFL, and MLC slabs using the Z project function. Each averaged image was cropped to only include averaged SVP, RNFL, and MLC slabs.

### 2.3. Identification and Quantification of MLCs

Two masked graders (SH and SMR) identified MLCs from averaged images using our previously published semi-automated custom macro in FIJI [20]. Briefly, this process reduced background noise, removed background irregularities, enhanced signal, binarized the image, and extracted discrete cell shapes of potential MLCs. Graders manually marked vessel artifacts and other irregularities to be removed. Additionally, graders removed unwanted noise and added MLCs that were not automatically identified. We calculated MLC density (count per image area) and MLC percent image area using the Analyze Particles function of FIJI. MLC density (r = 0.83, *p* < 0.0001) and MLC percent area (r = 0.82, *p* < 0.0001) showed high agreement between graders. MLC density and MLC percent area were averaged between graders for statistical analysis between control and RVO eyes.

### 2.4. Calculation of Perifoveal SVP Density

Averaged SVP slabs were binarized using the Thresholding function (Moments algorithm) in FIJI. The threshold level was adjusted so that vessels were as visible as possible without any visible background irregularities. A 1 mm diameter circle centered on the fovea was used to delete the fovea and parafoveal region from analysis. The area of binarized vessels was measured and presented as a ratio of the measured image area. 

### 2.5. Statistical Analysis

MLC density, MLC percent area, SVP density, BCVA at initial visit, CST at initial visit, and number of intravitreal injections at the initial visit were tested for normality using the Anderson-Darling, D’Agostino and Pearson, Shapiro–Wilk, and Kolmogorov–Smirnov tests. In the majority of tests, MLC density, MLC percent area, and SVP density were normally distributed while BCVA, CST, and number of intravitreal injections were not normally distributed. Pearson’s correlation was used to compare MLC metrics between graders. Due to excellent agreement in MLC density (r = 0.83, *p* < 0.0001) and MLC percent area (0.82, *p* < 0.0001) between graders, MLC parameters between graders were averaged. MLC density, MLC percent area, and SVP density were compared between control and RVO eyes using Student’s paired t-test between control and RVO eyes of each subject. Non-parametric BCVA, CST, and number of intravitreal injections were compared between the control and RVO eyes using Wilcoxon matched-pairs rank test. Correlations between normally distributed MLC parameters and SVP were calculated using Pearson’s correlation. MLC measurements were correlated with continuous, parametric variables using Pearson’s correlation: SVP density and age. MLC metrics were correlated with categorical variables or non-parametric continuous variables using Spearman’s correlation: CST, BCVA, sex, RVO type, number of intravitreal injections, diabetes, and hypertension. 

Normality for number of intravitreal injections at 6 and 12 months, and BCVA at 12 months was tested using the Anderson-Darling, D’Agostino and Pearson, Shapiro–Wilk, and Kolmogorov–Smirnov tests. In the majority of tests, number of injections at 6 months was non-parametric, while the number of injections at 12 months was normally distributed. Correlations between MLC parameters and number of injections were made using Spearman’s correlation for 6-month data, and Pearson’s correlation for 12-month data. BCVA at 12 months was non-normally distributed in 2 of 4 tests, and Spearman’s correlation was used. 

## 3. Results

This prospective, cross-sectional OCTA imaging study included 23 unilateral RVO subjects. RVO eyes were compared to control eyes for all 23 participants. The study population had a mean age of 66 years, were 52% (12 of 23) female, and was comprised of 52% (12 of 23) BRVO and 48% (11 of 23) CRVO eyes (Table 1). Comorbid conditions included diabetes in 7 of 23 (30%) and hypertension in 18 of 23 (78%) participants (Table 1). Patients with RVO demonstrated worse BCVA (20/42 vs. 20/23, *p* = 0.0008), a trend toward greater CST (295.5 + 183.4 vs. 228.6 + 30.4, *p* = 0.34), and had received more intravitreal injections (9 + 12.2 vs. 0, *p* < 0.0001, Table 1).

Figure 1 shows a representative example of a control (Figure 1A,B) and RVO (Figure 1C,D) eye with MLCs identified in red (Figure 1B,D). Agreement between graders was high for both MLC density (r = 0.83, *p* < 0.0001, Figure 1E) and MLC percent area (r = 0.82, *p* < 0.0001, Figure 1F), demonstrating the reproducibility of our methodology. We found that RVO eyes demonstrated a 1.86-fold increase in MLC density (*p* = 0.0266, Figure 1G), and 1.94-fold greater (*p* = 0.0415, Figure 1H) MLC percent area compared to control eyes.

Since we previously published that MLCs are increased in PDR patients who have significant macular ischemia [20], we measured superficial SVP density in our RVO subjects. SVP density was variable in our cross-sectional population, including RVO eyes with high (Figure 2A), medium (Figure 2B), and low SVP densities (Figure 2C). Overall, average SVP density was reduced from 0.4531 in control eyes to 0.3414 (25% reduction, *p* < 0.0001, Figure 2D) in RVO eyes. However, we found no significant correlation between SVP density and MLC density (r = −0.15, *p* = 0.33, Figure 2E) or MLC percent area (r = −0.25, *p* = 0.11, Figure 2F).

Since, control and RVO groups demonstrated significantly different BCVA, CST, and number of injections prior to imaging, we performed a univariate correlation analysis between MLC metrics and baseline demographics, clinical characteristics, ocular findings, and CST in all eyes (Table 2). MLC percent area and density were significantly correlated with each other (r = 0.752, *p* < 0.001). However, MLC density and percent area showed no significant correlation with baseline SVP density, CST, BCVA, age, sex, presence of diabetes, presence of hypertension, number of intravitreal injections prior to MLC imaging, or RVO type. These data suggest that MLCs have no correlation with baseline factors in this cross-sectional analysis.

Next, we investigated whether MLC percent area or density were correlated with the number of intravitreal injections at 6 and 12 months follow-up. Subjects who did not get any injections at Month 12 were excluded from this analysis. MLC density was positively correlated with the number of injections at 6 months (n = 12, r = 0.62, *p* = 0.03, Figure 3A) and 12 months (n = 9, r = 0.80, *p* = 0.009, Figure 3C) post-imaging. However, MLC percent area percent was not correlated with injection number at Month 6 (n = 12, r = 0.28, *p* = 0.37, Figure 3B) but did show a trend at Month 12 (n = 9, r = 0.60, *p* = 0.09, Figure 3D) post-imaging. 

Since MLC density and percent area were correlated with the number of future intravitreal injections at 12 months, we investigated whether MLC metrics were correlated with BCVA change from 0 to 12 months in all patients. Change in BCVA was measured by subtracting the LogMAR BCVA at the initial visit (0M) from the LogMAR BCVA at 12 months post-imaging (12M); thus, a negative change in BCVA indicates vision improvement. While MLC density did not correlate with BCVA change (n = 17, r = 0.25, *p* = 0.32, Figure 4A), MLC percent area was significantly correlated with BCVA change over 12 months (n = 17, r = 0.57, *p* = 0.02, Figure 4B). These data suggest that eyes with lower MLC percent area were correlated with BCVA improvement. 

## 4. Discussion

Since MLCs have been linked to retinal diseases characterized by ischemia and edema, such as PDR, we aimed to analyze MLCs in RVO patients. We found that MLC density and percent area were significantly higher in RVO eyes compared to control eyes (Figure 1). We found no correlation between MLC parameters and baseline characteristics including BCVA, age, sex, presence of diabetes, presence of hypertension, number of intravitreal injections prior to MLC imaging, CST, SVP, or RVO type (Figure 2, Table 2). However, MLC density was positively correlated with the number of intravitreal injections at 6 and 12 months post-imaging (Figure 3), and MLC percent area was positively correlated with BCVA change at 12 months (Figure 4). These data indicate that RVO eyes with more MLCs, were more difficult to treat, required more intravitreal injections, and had worse visual acuity outcomes. 

To our knowledge, Zeng et al. and Wang et al. are the only other published studies that investigated MLCs in RVO eyes [21,22]. Our results support those studies in that we also found MLC density and percent area were significantly higher in RVO eyes compared to control eyes. Additionally, in agreement, Zeng et al. show no association between MLC density and macular vessel density [21]. However, Zeng et al. did find that MLC density significantly correlated with increased macular thickness, which we did not observe [21]. Similarly, Wang et al. observed a positive correlation between MLC density and macular thickness in chronic eyes, but a negative association in acute eyes [22]. Since our study investigated a cross-sectional population including both acute and chronic eyes, we did not find an association between MLCs and baseline CST because the acute and chronic patients have inverse associations, canceling out each another.

MLCs are a heterogeneous group of macrophages, which include microglia, hyalocytes, and perivascular macrophages at steady-state, and additionally include monocytes and monocyte-derived macrophages during inflammation [18]. In experimental RVO in mice, microglia are activated, monocytes are recruited to the retinal tissue, and monocytes differentiate into monocyte-derived macrophages [19]. Complete monocyte depletion with clodronated liposomes increases retinal venous endothelial cell apoptosis while classical monocyte depletion alone has no effect upon endothelial cell apoptosis [23]. These data suggest that non-classical monocyte recruitment to tissue is protective for endothelial cells and may prevent ischemia during RVO. However, chronic microglia, monocyte, and macrophage depletion with a colony-stimulating factor receptor 1 antagonist led to diminished proinflammatory cytokines, improved retinal ganglion cell survival, and delayed retinal degeneration [24]. These data suggest that either microglia activation or blood-derived monocytes/macrophages are harmful to the neural retina during RVO. Our data that patients with less MLCs have improved visual acuity outcomes suggest that a portion of MLCs could be neurodegenerative activated microglia or blood-derived monocytes.

Hyalocytes are tissue resident macrophages of the cortical vitreous. Hyalocytes have not been investigated during RVO but are known to have phagocytic functions, antigen-presenting capabilities, and help modulate intraocular inflammation [25]. Furthermore, under hypoxic and inflammatory conditions, cultured hyalocytes have been shown to increase expression of VEGF [26]. Hyalocytes also aid in the synthesis of the extracellular matrix and have contractile properties, which in turn can contribute to epiretinal membrane, proliferative vitreoretinopathy, and retinal detachment [27]. Based upon these known hyalocyte functions, increased hyalocytes at the vitreoretinal interface in RVO patients could be an explanation for why patients with more MLCs required more intravitreal injections.

The main limitations to our study were its cross-sectional nature and moderate sample size. In agreement with Zeng et al, MLCs were increased in patients with RVO in our cohort despite our sample size of 23 per group compared to their 36 per group [21]. This moderate sample size and cross-sectional nature were the likely reasons why we found no significant correlations between MLC metrics and baseline factors. Furthermore, our sample size was further reduced in our analysis of patients requiring intravitreal injections, which included only 12 patients at 6 months and 9 patients at 12 months. Nevertheless, we still detected more intravitreal injections in patients with greater MLC numbers. It will be important to confirm this important finding in follow up studies.

## 5. Conclusions

We found that RVO eyes had significantly higher MLC density and percent area compared to control eyes. We also found that MLC parameters correlated with more future intravitreal injections and worsened vision acuity over 12 months. These data suggest that MLCs could be a biomarker for treatment resistant RVO eyes. Future studies are needed to determine how anti-inflammatory therapies like intravitreal steroids impact MLC numbers.

## Figures and Tables

**Figure 1 jpm-13-00045-f001:**
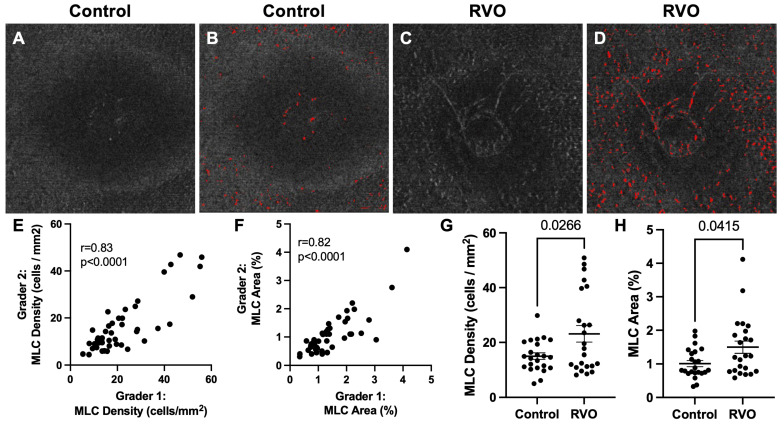
**MLCs are increased in RVO eyes.** (**A**,**B**) Representative image of a control eye (**A**) with marked MLCs in red (**B**). (**C**,**D**) Representative image of an RVO eye (**C**) with marked MLCs in red (**D**). Scatter plots demonstrating inter-rater reliability between two graders for MLC density (**E**) and MLC percent area (**F**). MLC density (**G**) and MLC percent area (**H**) were greater in RVO compared to control eyes.

**Figure 2 jpm-13-00045-f002:**
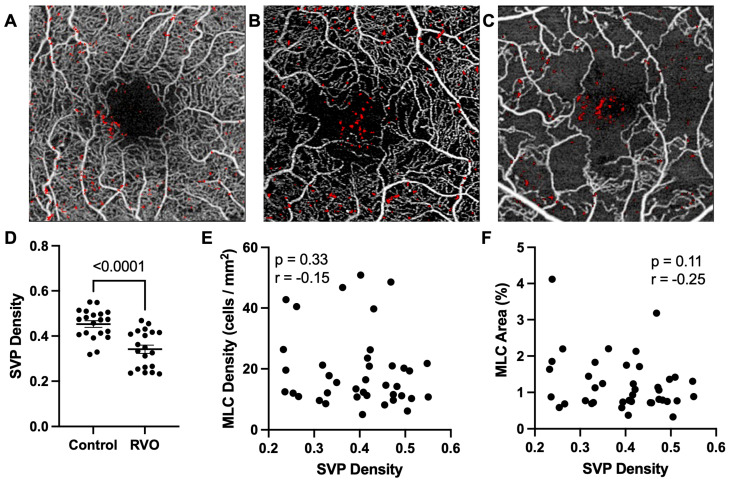
**MLCs do not correlate with SVP.** Representative examples of high (**A**), medium (**B**) and low (**C**) SVP density eye with MLCs marked in red. (**D**) Perifoveal SVP density was significant reduced in RVO eyes. MLC density (**E**) and MLC percent area (**F**) were not correlated with SVP density.

**Figure 3 jpm-13-00045-f003:**
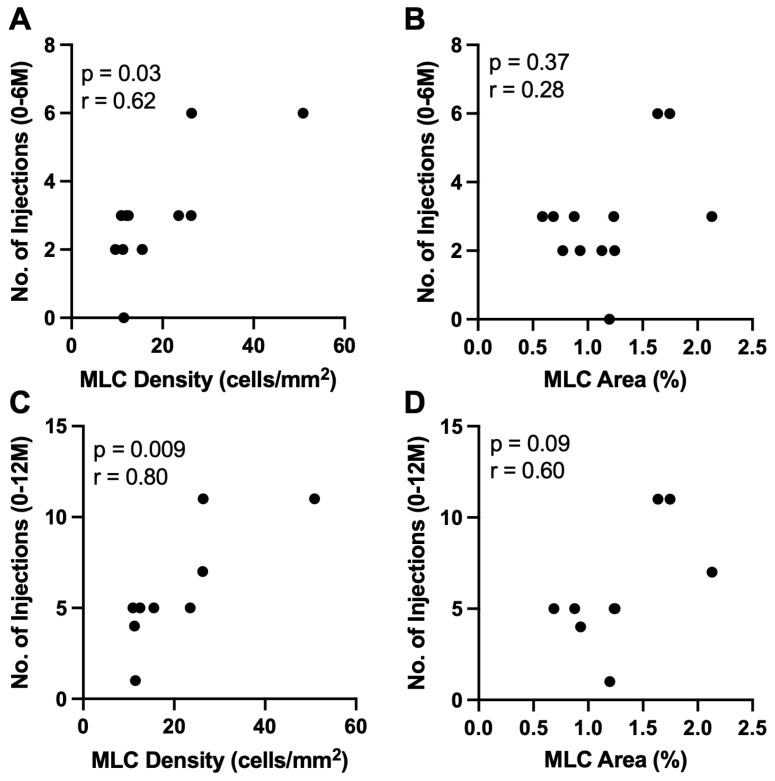
**MLC density correlates with number of future intravitreal injections.** MLC density (**A**,**C**) and MLC percent area (**B**,**D**) correlation with the number of intravitreal injections at 6 months (**A**,**B**) and 12 months post-imaging (**C**,**D**).

**Figure 4 jpm-13-00045-f004:**
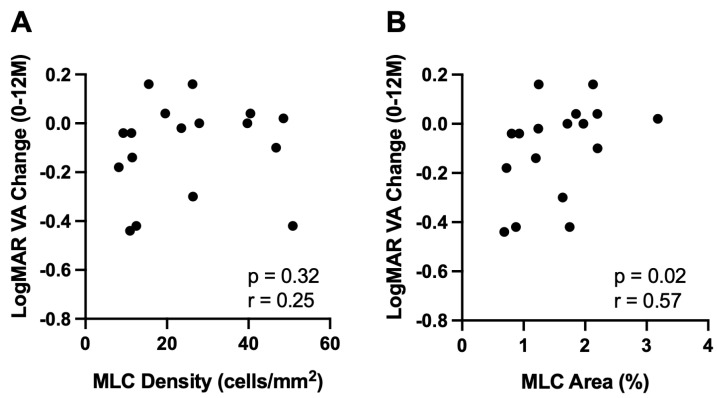
**MLC percent area correlated with worse BCVA over 12 months.** MLC density (**A**) and percent area (**B**) correlation with change in LogMAR BCVA over 12 months (12M BCVA–0M BCVA, thus negative numbers indicate BCVA improvement).

**Table 1 jpm-13-00045-t001:** Demographics Characteristics of Patients.

Patient Characteristic	Value	*p-Value*
Subjects, *n*	23	
Age (y), mean ± SD	66.2 ± 12.0	
Sex (female), *n* (%)	12 (52)	
RVO type		
BRVO, *n* (%)	12 (52)	
CRVO, *n* (%)	11 (48)	
DM, *n* (%)	7 (30)	
HTN, *n* (%)	18 (78)	
BCVA at initial visit, Snellen (LogMAR)		
Control Eye	20/23 (0.06)	
RVO Eye	20/42 (0.32)	0.0008
CST (μm), mean ± SD		
Control Eye	228.6 ± 30.4	
RVO Eye	295.5 ± 183.4	0.3408
Number of Injections, mean ± SD Control Eye RVO Eye	0 ± 0 9 ± 12.2	<0.0001

SD = standard deviation, RVO = retinal vein occlusion, BRVO = branch retinal vein occlusion, CRVO = central retinal vein occlusion, DM = diabetes mellitus, HTN = hypertension, BCVA = best-corrected visual acuity, CST = central subfield thickness. Comparisons were made using Wilcoxon matched-pairs rank test.

**Table 2 jpm-13-00045-t002:** Univariate correlation analysis between MLC and patient parameters for all eyes.

	MLC Percent Area r (*p*)	MLC Density r (*p*)
MLC Area	--	0.752 (*p* < 0.001)
SVP density	−0.257 (*p* = 0.110)	−0.157 (*p* = 0.332)
CST *	−0.057 (*p* = 0.708)	−0.156 (*p* = 0.996)
LogMAR BCVA *	0.028 (*p* = 0.855)	0.059 (*p* = 0.699)
Age	0.044 (*p* = 0.771)	−0.139 (*p* = 0.356)
Sex *	0.046 (*p* = 0.762)	−0.036 (*p* = 0.816)
DM *	−0.107 (*p* = 0.480)	0.014 (*p* = 0.925)
HTN *	−0.099 (*p* = 0.512)	0.143 (*p* = 0.343)
Number of Injections *	−0.251 (*p* = 0.249)	−0.288 (*p* = 0.183)
RVO type *	−0.013 (*p* = 0.953)	0.249 (*p* = 0.251)

Pearson Correlation was performed for continuous, parametric variables. Spearman Correlation (*) for categorical and continuous, non-parametric variables. Correlation coefficients are shown with *p*-values in parentheses. MLC = macrophage-like cell, SVP = superficial vascular plexus, CST = central subfield thickness, BCVA = best-corrected visual acuity, DM = diabetes mellitus, HTN = hypertension, RVO = retinal vein occlusion.

## Data Availability

Original data can be made available upon reasonable request to the corresponding author.

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
