# Peer review of "Macrophage-like Cells Are Increased in Retinal Vein Occlusion and Correlate with More Intravitreal Injections and Worse Visual Acuity Outcomes"

_jpm, 2022, doi:10.3390/jpm13010045_

Round 1

Reviewer 1 Report

The authors have used both parametric and non-parametric tests; they should provide details if they have performed any normality tests.

Since age and gender could significantly influence the MLC formation, have authors statistically adjusted for age and gender during correlation analysis? Although they did not find any significant correlation with age, sex, the presence of diabetes, or hypertension, since the small sample size was used for the study, it is important to use these parameters for the statistical adjustment.

The authors state that they did not find a significant association between SVP density and MLC density (Figure 2E ). Using correlation rather than association would be more appropriate as they have not performed regression analysis. A similar interpretation can be observed with other figures as well.

It needs to be made clear how did they predict the number of future intravitreal injections? Also, correlation analysis is not an appropriate method for predicting the prediction of future events.

Author Response

The authors have used both parametric and non-parametric tests; they should provide details if they have performed any normality tests.

Response: Normality testing was performed and the results were described on Line 129-132. Additional normality testing for 6 month and 12 month post-imaging results were discussed on Line 145-149.

Since age and gender could significantly influence the MLC formation, have authors statistically adjusted for age and gender during correlation analysis? Although they did not find any significant correlation with age, sex, the presence of diabetes, or hypertension, since the small sample size was used for the study, it is important to use these parameters for the statistical adjustment.

Response: We thank the reviewer for their thorough review. Importantly, our analysis was paired between RVO eye and fellow eye. Therefore, age and sex are completely accounted for when comparing MLC numbers between RVO eyes and fellow eyes. In correlation analysis, we did not adjust for age or sex because no correlation was detected. Furthermore, the correlation coefficients were 0.044, 0.046, 0.139, and 0.036 suggesting that age and sex had almost no association with MLC density or percent area. Therefore, adjusting for age and sex would have almost no effect upon the data. In corroboration, we similarly found no significant correlation between MLCs and sex in our recent paper published in Diagnostics (PMID: 36428853) on MLCs in patients with vision threatening diabetic retinopathy. Age was correlated with MLCs in univariate analysis but had no association with MLCs in multivariate analysis.

The authors state that they did not find a significant association between SVP density and MLC density (Figure 2E ). Using correlation rather than association would be more appropriate as they have not performed regression analysis. A similar interpretation can be observed with other figures as well.

Response: We did perform a correlation analysis. We wrote association to not be redundant in our writing. However, for clarity, as the reviewer points out, we have changed the results section on line 189 to state correlation. We made similar changes on Line 204, 271, 278 and 19-20 in the abstract.

It needs to be made clear how did they predict the number of future intravitreal injections? Also, correlation analysis is not an appropriate method for predicting the prediction of future events.

Response: We apologize for the lack of clarity. This was not a predictive model but rather a prospective analysis to determine if MLC metrics were correlated with the number of intravitreal injections at Month 6 and Month 12 post-imaging. We have changed the wording of the results section in Line 231-237 and Line 269 to improve clarity. We have also removed any mention of “future” or “prediction” in the manuscript in order to tone down our conclusions.

Reviewer 2 Report

In this manuscript, the authors describe how macrophages-like cells could be used as an indicator biomarker to study the severity, possible treatment, and possible outcome of this treatment in retinal vein occlusion patients.

Overall, the paper is well written, it is easy to read, and the data is well presented. However, there are a one important concern that the authors should address.

In the title and in the manuscript (Lines: 228-234) the author suggest that exist a correlation between the macrophages-like cells density and the number of the injection performed to the patient but does not clearly explain how they reached to the conclusion that through this relationship they can predict the number of injections that each patient will need as a treatment.

This is a fundamental cornerstone of the work and for this reason, the authors should explain it extensively.

Author Response

In this manuscript, the authors describe how macrophages-like cells could be used as an indicator biomarker to study the severity, possible treatment, and possible outcome of this treatment in retinal vein occlusion patients. Overall, the paper is well written, it is easy to read, and the data is well presented. However, there are a one important concern that the authors should address.

Response: We thank the reviewer for their time and effort reviewing this study.

In the title and in the manuscript (Lines: 228-234) the author suggest that exist a correlation between the macrophages-like cells density and the number of the injection performed to the patient but does not clearly explain how they reached to the conclusion that through this relationship they can predict the number of injections that each patient will need as a treatment. This is a fundamental cornerstone of the work and for this reason, the authors should explain it extensively.

Response: We apologize for the lack of clarity. This was not a predictive model but rather a prospective analysis to determine if MLC metrics were correlated with the number of intravitreal injections at Month 6 and Month 12 post-imaging. We have changed the wording of the results section in Line 231-237 and Line 269 to improve clarity. We have also removed any mention of “future” or “prediction” in the manuscript in order to tone down our conclusions.